# Kambin’s Triangle Approach versus Traditional Safe Triangle Approach for Percutaneous Transforaminal Epidural Adhesiolysis Using an Inflatable Balloon Catheter: A Pilot Study

**DOI:** 10.3390/jcm8111996

**Published:** 2019-11-15

**Authors:** Ho Young Gil, Sangmin Jeong, Hyunwook Cho, Eunjoo Choi, Francis Sahngun Nahm, Pyung-Bok Lee

**Affiliations:** 1Department of Anesthesiology and Pain Medicine, Anesthesia and Pain Research Institute, Ajou University College of Medicine, Suwon 16499, Korea; kilhoyoung@naver.com; 2Department of Anesthesiology and Pain Medicine, Multidisciplinary Pain Center, Seoul National University Bundang Hospital, Seongnam 13496, Korea; doctorsfitnessjeong@gmail.com (S.J.); zanggu@gmail.com (H.C.); ejchoi7956@gmail.com (E.C.); hiitsme@hanmail.net (F.S.N.)

**Keywords:** balloon, epidural adhesiolysis, Kambin’s triangle, low back pain, lumbar radicular pain, lumbar spinal stenosis, safe triangle, transforaminal, ZiNeu catheter

## Abstract

Spinal stenosis is a common condition in elderly individuals. Many patients are unresponsive to the conventional treatment. If the transforaminal epidural block does not exert a sufficient treatment effect, percutaneous transforaminal epidural adhesiolysis (PTFA) through the safe-triangle approach using an inflatable balloon catheter can reduce the patients’ pain and improve their functional capacity. We aimed to evaluate the safety and efficacy of the Kambin’s-triangle approach for PTFA using an inflatable balloon catheter and compare this approach to the traditional safe-triangle approach. Thirty patients with chronic unilateral L5 radiculopathy were divided into two groups: the safe-triangle-approach and Kambin’s-triangle-approach groups, with 15 patients each. The success rate of the procedure was assessed. Pain and dysfunction were assessed using the Numerical Rating Scale and Oswestry Disability Index, respectively, before the procedure and at 1 and 3 months after the procedure. The success rate of the procedure was high in both the groups, with no significant difference between the groups. The Numerical Rating Scale and Oswestry Disability Index scores significantly decreased 3 months after the procedure in both the groups, with no significant difference between the groups. For patients in whom the safe-triangle approach for PTFA is difficult, the Kambin’s-triangle approach could be an alternative.

## 1. Introduction

Spinal stenosis is a common condition in elderly individuals, causing lower back and radicular pain. Many of these patients are unresponsive to conservative treatment, such as physical therapy, non-steroidal anti-inflammatory drugs, and muscle relaxants [1,2]. For these patients, epidural steroid injections (ESIs) are considered, which are effective because of their anti-inflammatory, membrane stabilizing, and antihyperalgesic effects. Local anesthetics alter or interrupt the nociceptive input, reflex mechanism of the afferent fibers, self-sustaining activity of the neurons, and pattern of neuronal activities. Additionally, hyaluronidase has been used to enhance the lysis of epidural adhesions [3]. However, data on the benefits of ESI are conflicting, and most studies report only short-term benefits [4]. The long-term effects are also positive, however, they are of a smaller magnitude and not statistically significant [1]. In these cases, the epidural space is restricted by perineural or epidural adhesions/fibrotic tissues, and the injectate frequently fails to spread effectively into the ventral epidural space [5].

Percutaneous epidural adhesiolysis (PEA) is a minimally invasive procedure in which a catheter is placed directly into the adhesions or scar tissue compressing the nerve root. PEA can eliminate the effects of adhesions and ensure the delivery of high concentrations of injected drugs to target areas [5]. A high-quality randomized controlled trial showed fair evidence [1]. However, it is a useful treatment method for chronic pain refractory to other treatments [6,7].

Recently, a randomized controlled trial involving cases of unsuccessful resolution of adhesions or insufficient relief of stenosis with conventional methods demonstrated that PEA using an inflatable balloon catheter resulted in significantly higher pain reduction and functional improvement for 6 months compared with PEA using the Racz catheter (estimated difference (95% confidence interval), for back pain −2.02 (−3.58 to −0.45), for leg pain −1.88 (−3.15 to −0.61), and for Oswestry Disability Index (ODI) −13.74 (−22.18 to −5.30)) [8].

Efficacy of the transforaminal approach has been shown in patients refractory to conventional PEA using the caudal approach [9]. The transforaminal approach is preferred for ESI because drugs can be injected in the anterior epidural space, and drug concentrations around the relevant nerve root can be maximized [10]. Moreover, the addition of hypertonic saline to transforaminal ESIs results in superior short-term pain-relieving efficacy [11]. This can be explained by adhesiolysis and the possible neuromodulatory effects of hypertonic saline. Furthermore, percutaneous transforaminal epidural adhesiolysis (PTFA) with an inflatable balloon catheter using the safe-triangle approach can reduce patients’ pain and improve their functional capacity [12,13].

However, the safe-triangle approach may cause irritation of the spinal nerve root during the injection in cases of severe spinal stenosis, epidural fibrosis, and intervertebral disk degeneration. An alternative transforaminal approach is the Kambin’s-triangle approach, in which the needle is located in the inferior–posterior or lateral view, reducing the risk of irritation of the spinal nerve root. The safety and efficacy of the Kambin’s-triangle approach for transforaminal epidural injection have been demonstrated [14].

In this study, we aimed to investigate the safety and efficacy of the Kambin’s-triangle approach for PTFA using an inflatable balloon catheter and compare this approach to the traditional safe-triangle approach.

## 2. Materials and Methods

### 2.1. Patients

This was a prospective, randomized, controlled study. This study was approved by the Institutional Review Board of Seoul National University Bundang Hospital (No. B-1708/415-304) in September 2019. Patient recruitment and data collection were performed from September 28, 2017 to March 28, 2018. All participants were provided with written and verbal information on the trial before obtaining their written consent. The inclusion criteria were as follows: (1) age 20–80 years; (2) unilateral L5 radiculopathy for more than 3 months; (3) no symptom improvement or continuation of pain relief for more than 1 month after physical therapy and medication; and (4) no or <50% pain reduction in response to transforaminal epidural injection. The exclusion criteria were as follows: (1) unilateral L5 radiculopathy for less than 3 months; (2) prior lumbar surgery; (3) occurrence of adverse effects due to local anesthetics, contrast medium, or steroids; (4) hemostatic disorders; (5) infections, cancer, severe neurological defects, or cauda equina syndrome; and (6) uncontrollable medical or psychiatric illness.

A total of 30 patients with lumbar spinal stenosis were included in this study; the safe-triangle-approach and Kambin’s-triangle-approach groups comprised 15 patients each. Two patients from each group dropped out. Two patients in the safe-triangle-approach group were excluded: one decided to opt out of the study, and the other was lost to follow-up. Two patients from the Kambin’s-triangle-approach group were excluded as they were lost to follow-up (Table 1).

### 2.2. Study Design and Randomization

A random allocation software was used to randomly allocate patients into two groups. Ultimately, the safe-triangle-approach and Kambin’s-triangle-approach groups comprised 15 patients each. This was a double-blinded study in which the experimenters and participants were blinded to the group allocation. All patients underwent PTFA using an inflatable balloon catheter.

### 2.3. Percutaneous Transforaminal Epidural Adhesiolysis Using an Inflatable Balloon Catheter

PTFA was performed using an inflatable balloon catheter with the safe-triangle approach or Kambin’s-triangle approach under fluoroscopy, by an experienced pain physician, who was blinded to the purpose and design of this study. In the traditional safe-triangle approach, a 16-gauge Tuohy needle (JUVENUI Co., Ltd., Seongnam, Republic of Korea) was inserted in the intervertebral foramen between L5 and S1 (just below the L5 pedicle). Fluoroscopy and contrast medium injection were used to check that the end of the needle had entered the anterior epidural space. In the Kambin’s-triangle approach, a 16-gauge Tuohy needle (JUVENUI Co., Ltd.) was inserted just above the superior articular process of L5 under oblique fluoroscopy (Figure 1 and Figure 2). In both cases, when the needle was positioned in the anterior epidural space, a 2-French inflatable balloon catheter (ZiNeu F: JVN_FC01, JUVENUI Co., Ltd., Seongnam, Republic of Korea) was inserted in the Tuohy needle. The correct positioning of the end of the balloon catheter in the narrowed epidural space around the L5 nerve root was verified. After the catheter had entered the target point, the tip of the Tuohy needle was retracted outside the intervertebral foramen to prevent loss and damage of the catheter and balloon. Inflation and deflation of the balloon from inside the lateral recess to outside the intervertebral foramen were repeated at a minimum of five consecutive points. Each balloon session lasted less than 5 s. The catheter was pre-filled with contrast media, and the maximal inflated balloon diameter was determined within 6 mm upon injecting 0.13 mL of contrast media. This inflation was repeated three times at each inflation point, considering the patient’s response. After the termination of inflation, the deflated balloon catheter was removed. Subsequently, the Tuohy needle was inserted in the anterior epidural space under fluoroscopy, and a 3-mL mixture of 0.18% ropivacaine, 1500 IU hyaluronidase, and 5 mg dexamethasone was injected.

### 2.4. Evaluation of Outcome Variables

The success of the procedure, pre- and post-procedure pain, and the degree of dysfunction were evaluated by an independent physician who was blinded to the study design and group assignment. The success of the procedure was assessed using three categories (B, D, and F): in category B (ballooning), the device entered the target area, and the contrast medium diffusion was successful after balloon inflation; in category D (dye spread), balloon inflation failed, but contrast medium diffusion and adhesiolysis were successful; and in category F (fail), balloon inflation and adhesiolysis failed, so contrast medium diffusion could not be confirmed.

Patients’ degree of pain was measured using a standardized 11-point (0–10) Numerical Rating Scale (NRS) and evaluated by a well-trained physician at baseline and 1 and 3 months after the procedure. The severity of pain was scored from 0 to 10, where “0” represented no pain at all, and “10” represented the worst possible pain. The patients were encouraged to express their feelings regarding the pain.

The ODI was used to assess the patients’ degree of dysfunction. ODI assessments were performed at baseline and 1 and 3 months after the procedure. ODI is a 10-item questionnaire used globally to functionally assess patients with low back pain [15]. However, in this study, we used the 9-item Korean version of ODI, which excludes the assessment of sexual function for cultural reasons [16].

### 2.5. Statistical Analysis

All statistical analyses were performed using the SPSS Statistics program, version 21.0 (IBM Corp, Armonk, NY). Demographic data of the patients were analyzed using the chi-squared test and Fisher’s exact test. The success or failure of the procedure was analyzed using the Fisher’s exact test. In addition, a linear mixed model analysis was used to determine the differences in NRS and ODI scores at the following time points: before the procedure and immediately and 1 and 3 months after the procedure. A *p*-value < 0.05 was considered statistically significant.

## 3. Results

Regarding the success of the procedure itself, balloon inflation and adhesiolysis failed in two patients in the safe-triangle-approach group, and balloon inflation failed, but contrast medium diffusion and adhesiolysis were successful in one patient in the Kambin’s-triangle-approach group. The success rate of the procedure was high in both the groups, with no significant difference between the groups (safe-triangle approach: 76.92% vs. Kambin’s-triangle approach: 92.31%; *p* = 0.593, Fisher’s exact test) (Table 2).

Changes in the NRS score were analyzed using a linear mixed model. In both the groups, 1 month after the procedure, the NRS score decreased compared to that before the procedure, although without statistical significance. Furthermore, in both the groups, the NRS score significantly decreased 3 months after the procedure compared to that before the procedure, with no significant difference between the groups (Table 3 and Table 4).

Changes in the ODI score were also analyzed using a linear mixed model. Similarly, in both the groups, the ODI score decreased 1 month after the procedure compared to that before the procedure, although without statistical significance. Furthermore, in both the groups, the ODI score significantly decreased 3 months after the procedure compared to that before the procedure, with no significant difference between the groups (Table 3, Table 4).

The severity of pain during the procedure, which was assessed using NRS, was comparable between the two groups and did not differ significantly (safe-triangle-approach: 6.2 ± 2.74 vs. Kambin’s-triangle-approach: 6.2 ± 2.39; *p* = 0.782). There were no significant differences between the two groups in the number of cases of severe neurostimulation requiring cessation of the procedure or number of cases requiring needle readjustment because of injection of the contrast medium into a blood vessel.

## 4. Discussion

In the current study, there were no significant differences in pain, functional capacity, or the success rate of the procedure between the safe-triangle-approach and Kambin’s-triangle-approach groups. However, this is the first randomized trial to demonstrate the clinical efficacy of the Kambin’s-triangle approach for PTFA using an inflatable balloon catheter [17]. The results of this study suggest that the Kambin’s-triangle approach is as effective as the conventional safe-triangle approach. Therefore, the Kambin’s-triangle approach could be an alternative for patients in whom the safe-triangle approach for PTFA is difficult.

PEA is used in patients with refractory chronic low back and radicular pain or following failed back surgery syndrome [18]. The pain in these cases was found to occur not only in response to mechanical stimuli, but also due to chemical irritation around the nerve roots [19]. Additionally, leakage of the disc material into the epidural space can lead to inflammation and consequent epidural fibrosis, which result in compression of the nerve roots and tethering of the dura [20]. Therefore, the purpose of PEA is to eliminate problematic adhesions while enabling the targeted delivery of medications.

Conventional PEA may be divided into either chemical adhesiolysis using 10% hypertonic saline or mechanical adhesiolysis using a steerable catheter, such as NaviCath (Myelotec Inc, Roswell, GA). PEA performed with a wire-type catheter, such as the Racz catheter, is based on the concept of chemical adhesiolysis through the administration of medications to the target site. In contrast, a steerable catheter enables the physician to place the catheter tip near the nerve root for precise delivery of medications. Furthermore, it can facilitate mechanical adhesiolysis [6]. In addition to this, an inflatable balloon catheter can increase the diameter of the epidural space in the region of the intervertebral foramen by 10.5%–31.8% through inflation and deflation [12]. Therefore, PEA using the inflatable balloon catheter leads to significant pain reduction and functional improvement compared to PEA using a balloon-less catheter [8].

Although the aforementioned conventional PEA is frequently applied, in patients with pain refractory to conventional PEA, a procedure for more effective and safer PEA is needed. Therefore, PTFA was introduced, which demonstrated good results [9]. Additionally, PTFA using an inflatable balloon catheter leads to both significant pain relief and functional improvement in patients with refractory spinal stenosis [12].

The safe triangle is formed by the diagonal path of the nerve, base of the vertebral pedicle, and outer boundary of the vertebral body (Figure 1). This triangle is called the safe triangle because only the spinal nerves and blood vessels exist in this space [21]. While approaching the safe triangle, careful attention is required for the Adamkiewicz artery. In 75% of the cases, this artery passes through the intervertebral foramen from T9 to L1. However, in rare cases, it passes through the intervertebral foramen from L2 to L4 [22]. There have been several reports of spinal cord ischemia caused by invasion of the Adamkiewicz artery during the transforaminal nerve block [23,24]. A previous study using spinal angiography revealed that the Adamkiewicz artery is located in the superior half, lower third, and inferior fifth of the intervertebral foramen in 97%, 2%, and 0% of the cases, respectively [25]. Therefore, it is essential to verify that there is no vascular uptake of contrast medium before performing PTFA using an inflatable balloon catheter.

The Kambin’s triangle is formed by the path of the spinal nerve, upper border of the lower vertebral body, and inner boundary of the upper vertebral pedicle (Figure 1). The Kambin’s-triangle approach reduces nerve root stimulation and tissue damage around the nerves and prevents venous congestion, nerve edema, and epidural bleeding [26]. However, this approach carries the risk of injecting contrast medium into the disk, leading to complications, such as diskitis [27]. In this study, contrast medium injection into the disk or complications were not noted.

Either the safe-triangle or the Kambin’s-triangle approach may be used for transforaminal ESIs; both approaches showed similar effects on pain reduction at 2 months [28]. However, the possibility of spinal nerve irritating using the Kambin’s-triangle approach is small [14]. Additionally, in some patients, the epidural space cannot be accessed using the safe-triangle approach [29]. In cases of severe stenosis of the intervertebral foramen and epidural adhesions or severe degenerative changes in the disc, it is not easy to approach the stenotic site without irritating the nerve roots [27]; accessing the epidural space through the Kambin’s triangle could result in successful balloon catheter insertion, without irritation of the nerve root [17].

Several factors may contribute to pain relief and functional improvement noted after PEA using an inflatable balloon catheter. First, expansion of the epidural space by the balloon catheter may lead to effective mechanical adhesiolysis of perineural adhesions. Moreover, breaking or lengthening the thin transforaminal ligament can lead to expansion of the marginal space in the intervertebral foramen [12,30]. Second, inflation of the balloon catheter may decrease venous congestion, an essential factor inducing neurogenic claudication and circulatory disturbance in the nerve roots. Third, inflation of the balloon catheter may contribute to more effective drug delivery [12].

There are several limitations of this study. First, the total number of patients included in this study was relatively small, so a definitive conclusion could not be drawn. Second, the type and severity of stenosis, which may affect the success rate of the procedure, were not evaluated on imaging. Furthermore, the effect of the procedure may vary depending on whether the main cause of pain is the intervertebral disc or the bony intervertebral foramen. Third, there were a number of confounding variables, including the patient’s pain duration and functional status. We suspect that the age distribution of in our study showed a large variation and influenced the results, such as ODI. Fourth, this is not a double-blind study. This bias may affect the results; double-blind studies are needed in the future. Fifth, the patients were followed-up only for 3 months after the procedure; hence, only the short-term results were evaluated.

Further studies should be conducted to overcome the aforementioned limitations. These studies should include more patients, use detailed inclusion and exclusion criteria, include a wide variety of third variables, and have a long observation period.

In summary, as in the case of transforaminal ESIs, the Kambin’s-triangle approach could be an alternative for patients in whom the safe-triangle approach for PTFA is difficult. A randomized, controlled, double-blind, multi-center study should be conducted to support the findings of this study

## Figures and Tables

**Figure 1 jcm-08-01996-f001:**
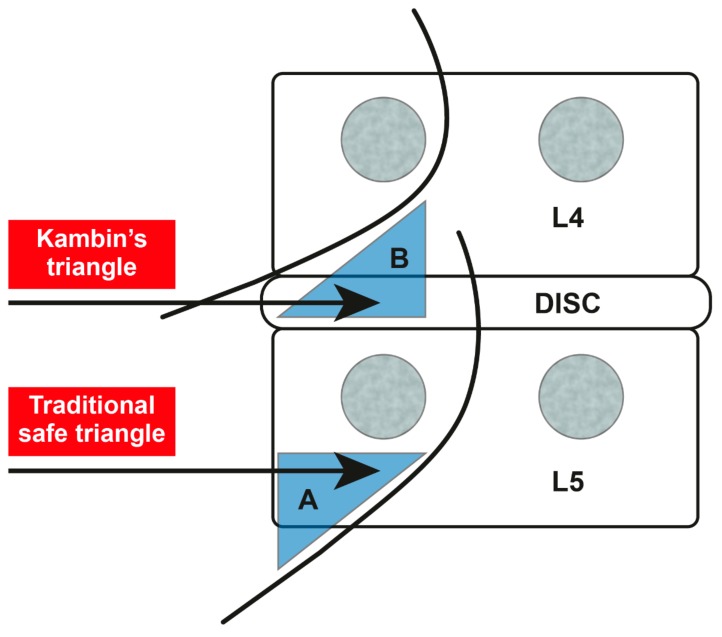
The safe-triangle approach (**A**) and the Kambin’s-triangle approach (**B**).

**Figure 2 jcm-08-01996-f002:**
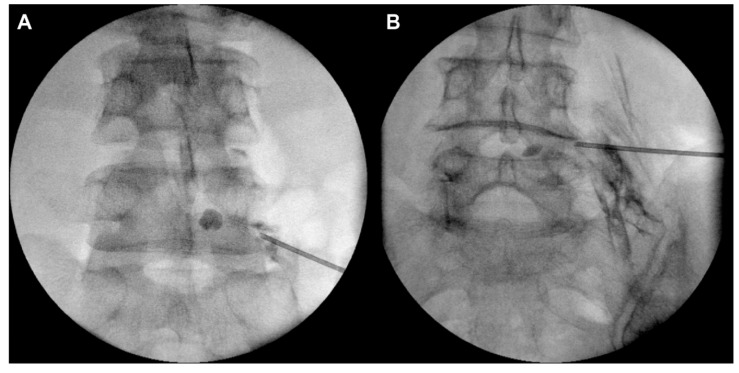
Fluoroscopic anterior-posterior view of the safe-triangle approach (**A**) and the Kambin’s-triangle approach (**B**).

**Table 1 jcm-08-01996-t001:** Characteristics of the patients.

Variables	Safe-Triangle-Approach Group (*n* = 12)	Kambin’s-Triangle-Approach Group (*n* = 12)
Age (years)	65.3 ± 21.7	68.0 ± 15.6
Sex		
Male	6 (46.1%)	5 (38.5%)
Female	7 (53.9%)	8 (61.5%)
Height (cm)	158.6 ± 14.4	160.6 ± 8.1
Weight (kg)	64.3 ± 16.5	62.6 ± 12.5
Body mass index (kg/m^2^)	25.2 ± 3.1	24.1 ± 3.6
Pain duration (years)	4.7 ± 4.6	10.1 ± 8.3
Target root		
Right L5	5 (38.5%)	9 (69.2%)
Left L5	8 (61.5%)	4 (30.8%)

Data are expressed as mean ± standard deviation for numerical variables.

**Table 2 jcm-08-01996-t002:** Comparison of the success rate of the procedure between the safe-triangle and Kambin’s-triangle approaches.

Approach	Success	Failure		Total	Success Rate	*p*-Value
	B^1^	D^2^	F^3^		B/(B + D + F) × 100	
Safe triangle	10	1	2	13	76.92%	
Kambin’s triangle	12	1	0	13	92.31%	
Total	22	2	2	26		0.593 *

* *p* > 0.05. Fisher’s exact test was used. ^1^ B: successful balloon inflation. ^2^ D: balloon inflation failed, but adhesiolysis and contrast medium diffusion were successful. ^3^ F: balloon inflation and adhesiolysis failed.

**Table 3 jcm-08-01996-t003:** Comparison of Numerical Rating Scale score and Oswestry Disability Index score before and after the procedure in the safe-triangle-approach and Kambin’s-triangle-approach groups.

Variables	Approach	Before Procedure	One Month After	Three Months After
Numerical Rating Scale	Safe triangle	6.0 ± 2.05	4.9 ± 2.02	4.0 ± 2.11 *
	Kambin’s triangle	5.3 ± 1.89	4.5 ± 2.12	4.0 ± 2.58 *
Oswestry Disability Index	Safe triangle	17.3 ± 6.48	12.6 ± 5.50	11.2 ± 4.92 *
	Kambin’s triangle	18.7 ± 8.38	18.1 ± 9.83	13.6 ± 7.37 *

Data are presented as mean ± standard deviation for numerical variables. * *p* < 0.05. Comparison before and after the procedure.

**Table 4 jcm-08-01996-t004:** An analysis of the Numerical Rating Scale and Oswestry Disability Index scores before and after the procedure in the safe-triangle-approach and Kambin’s-triangle-approach groups using a linear mixed model.

Variables		Coefficient	Standard Error	*p*-Value
**NRS**	Time			
	Before	Reference		
	One Month After (a)	−1.1	0.56	0.051
	Three Months After (b)	−2	0.56	<0.001*
	**Group**			
	Safe Triangle (1)	Reference		
	Kambin’s Triangle (2)	−0.7	0.91	0.441
	**Time × Group**			
	a 2	0.3	0.80	0.706
	b 2	0.7	0.80	0.379
**ODI**	Time			
	Before	Reference		
	One Month After (a)	−4.7	1.10	<0.001*
	Three Months After (b)	−6.1	1.10	<0.001*
	**Group**			
	Safe Triangle (1)	Reference		
	Kambin’s Triangle (2)	1.4	3.09	0.650
	**Time × Group**			
	a 2	4.1	1.56	0.009*
	b 2	1	1.56	0.521

* *p* < 0.05.

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
