# Peer review of "Kambin’s Triangle Approach versus Traditional Safe Triangle Approach for Percutaneous Transforaminal Epidural Adhesiolysis Using an Inflatable Balloon Catheter: A Pilot Study"

_jcm, 2019, doi:10.3390/jcm8111996_

Round 1
Reviewer 1 Report
GENERAL COMMENTS:
This study compared the effectiveness of the Kambin’s triangle approach versus the safe triangle approach for the percutaneous transforaminal epidural adhesiolysis (PTFA) using an inflatable balloon catheter. Although clinically relevant I consider there are major improvements in this paper to be performed. My main points for changes are described below, under the specific comments.
SPECIFIC COMMENTS:
Title: If both approaches are being compared in the study I suggest changing the title to "The Kambin’s Triangle versus traditional safe triangle approach for Percutaneous Transforaminal Epidural Adhesiolysis Using an Inflatable Balloon Catheter: A Pilot Study" for clarity.
Abstract
L15-16: "Recently, it was reported that percutaneous transforaminal epidural adhesiolysis (PTFA) using an inflatable balloon catheter can reduce patients’ pain and improve their functional capacity.". Briefly explain this clinical approach to improve background information of abstract.
L20: How were participants allocated to each group? A brief explanation is needed.
L23-24: Consider also presenting effect sizes as these can be important for this type of clinical comparison.
L27-28: Effect size results should be presented to support such conclusions.
Introduction
L40: Further explore with more details the conflicting results of these studies here. Why do most of them only present a short-term benefit?
L47: Same here, more details of the results of studies that have found this treatment method to be useful in patients with chronic pain should be explored. Do these studies also present only a short-term benefit?
L51: "Recently, a randomized controlled trial on cases where conventional methods fail to remove adhesions or sufficiently relieve stenosis demonstrated that PEA using an inflatable balloon catheter resulted in significant pain reduction and functional improvement". For how long do these improvements last? Explore results with some more details to improve rationale of why conducting this study.
L56-57: Explore reasons/mechanisms that explain these superior pain relieving effects.
L67: What were the hypothesis of the study? In this same sentence also explore specifically why this comparison is clinically important to be performed.
Methods
L73: How many men and women were included in the sample? Additionally, what were the physical/exercise weekly levels/capacity fo participants? More details regarding participants characteristics may be important to be included as the age range of 20-80 years is very large and could influence the clinical efficiency of both approaches. I would also like to see participants characteristics per group as potential differences between participants form each group could be a limitation of the study,
L103-109: These procedures need to be referenced by previous research utilizing similar methods.
L123: Precisely how long before and after the procedure were patients assessed for pain and ODI? Please add more details.
Statistical Analysis
L135-137: More details are needed in regards to the LMM approach used. For instance, did you run a LMM for repeated measures? What was and how was determined the covariance structure of the model(s) used? What specific variables were tested? All these details are needed when describing your statistics.
Results
L139: "3.1. Participants". This section should be moved up and added to the subheading "2.1 Patients" in the methods. Table 1 should also be moved to the methods for clarity.
L161-164: Indicate the % difference of the decreases in each time point compared to before the procedure. Additionally, although no significant differences were found between procedures it would be interesting to present effect sizes to better interpret the clinical value and efficiency of each procedure. Therefore, please add the effect sizes for NRS and ODI for each procedure on the different time points.
Discussion
L182-184: Please restate the aims and summarise results and conclusions in the first paragraph of the discussion for clarity. Also, if you did not find differences between both approaches why are you claiming that the Kambin's triangle approach is clinically efficient? Revise.
L182-194: The first 4 paragraphs of your discussion should added up and summarised in only one (1st) paragraph.
L229: The discussion sections needs to be improved by clearly exploring aims, results and conclusions of this and other studies presented/cited to enable proper comparisons to your results.
L231-232: "First, the total number of patients included in this study was relatively small, so the study lacked a clear conclusion." Do you mean that the study was underpowered? Did you perform any sample size calculation to avoid this issue?
234-235: "functional status". How did you control for this factor between participants of the sample? Additionally, do you consider that the large age range and potential difference on physical level and age between groups may have influenced results of the study? If yes, these points should also be discussed in the limitations paragraph as well.
L240: "In summary, the Kambin’s triangle approach could be an alternative option...". Revise. Since no significant difference was found between approaches why do you consider that the Kambin's triangle approach could be an alternative option compared to the safe triangle? Presenting non-significant % difference and effect size results between groups may improve such conclusions.
Author Response
Title: If both approaches are being compared in the study I suggest changing the title to "The Kambin’s Triangle versus traditional safe triangle approach for Percutaneous Transforaminal Epidural Adhesiolysis Using an Inflatable Balloon Catheter: A Pilot Study" for clarity.
- Revision Manuscript. Line number 2-5.
I agree with your opinion. I revised it.
Abstract
L15-16: "Recently, it was reported that percutaneous transforaminal epidural adhesiolysis (PTFA) using an inflatable balloon catheter can reduce patients’ pain and improve their functional capacity.". Briefly explain this clinical approach to improve background information of abstract.
- Revision Manuscript. Line number 16-19.
If the transforaminal epidural block does not have sufficient treatment effect, it was reported that percutaneous transforaminal epidural adhesiolysis (PTFA) through the safe triangle using an inflatable balloon catheter can reduce patients’ pain and improve their fucntional capacity.
L20: How were participants allocated to each group? A brief explanation is needed.
- Revision Manuscript. Line number 21-22.
Thirty patients with chronic unilateral L5 radiculopathy were divided into two groups of 15 patients each
L23-24: Consider also presenting effect sizes as these can be important for this type of clinical comparison.
- I agree with your opinion. But this is a pilot study. There is no prior information to estimate sample size. We calculated the number of subjects by referring to the most similar article (pain physician 2013; 16:213-224). Using two group t-test, for 80% power and a two-sided significance level of 0.05, 8 patients were required in each study group, for a total sample size of 16 participants. Assuming 10% dropout, 9 patients were required in each group. Assuming 25% dropout, 10 patients were required in each group. The effect size is estimated at 1.615.
So we referenced the article of Whitehead et al. (Stat Methods Med Res. 2016 Jun; 25(3): 1057–1073.) According to their opinion, for 80% power and smaller standardized effect size, 20 patients were required. Assuming 25% dropout, 13 patients were required in each group.
We also contacted the hospital statistics team. They replied that it was okay to proceed because it was a pilot study.
L27-28: Effect size results should be presented to support such conclusions.
- I agree with your opinion. I answered above.
Introduction
L40: Further explore with more details the conflicting results of these studies here. Why do most of them only present a short-term benefit?
- Revision Manuscript. Line number 43-44.
The long-term effects were also positive; however, they were of smaller size and not statistically significant.
L47: Same here, more details of the results of studies that have found this treatment method to be useful in patients with chronic pain should be explored. Do these studies also present only a short-term benefit?
- Revision Manuscript. Line number 49.
A high-quality randomized controlled trial showed fair evidence. However, it is a useful treatment method for chronic pain refractory to other treatments.
L51: "Recently, a randomized controlled trial on cases where conventional methods fail to remove adhesions or sufficiently relieve stenosis demonstrated that PEA using an inflatable balloon catheter resulted in significant pain reduction and functional improvement". For how long do these improvements last? Explore results with some more details to improve rationale of why conducting this study.
- Revision Manuscript. Line number 51-54.
Recently, a randomized controlled trial on cases of failure to remove adhesions or sufficiently relieve stenosis with conventional methods demonstrated that PEA using an inflatable balloon catheter resulted in significantly higher pain reduction and functional improvement for 6 months compared to PEA using Racz catheter.
L56-57: Explore reasons/mechanisms that explain these superior pain relieving effects.
- Revision Manuscript. Line number 60.
This can be explained by adhesiolysis and the possible neuromodulatory effects of hypertonic saline.
L67: What were the hypothesis of the study? In this same sentence also explore specifically why this comparison is clinically important to be performed.
- Revision Manuscript. Line number 69-71.
In this study, we aimed to investigate the safety and efficacy of the Kambin’s triangle approach for PTFA using an inflatable balloon catheter and to compare this approach to the traditional safe triangle approach.
Methods
L73: How many men and women were included in the sample? Additionally, what were the physical/exercise weekly levels/capacity fo participants? More details regarding participants characteristics may be important to be included as the age range of 20-80 years is very large and could influence the clinical efficiency of both approaches. I would also like to see participants characteristics per group as potential differences between participants form each group could be a limitation of the study,
- Revision Manuscript. Line number 224-227.
I agree with your opinion. So I presented as a limitations.
Third, a number of confounding variables exist, including the patient’s pain duration and functional status. We suspect that the age distribution of in our study showed a large variation and influenced the results, such as ODI.
L103-109: These procedures need to be referenced by previous research utilizing similar methods.
- We referenced several papers on the PEA procedure using balloon catheter.
Karm, M.H.; Choi, S.S.; Kim, D.H.; Park, J.Y.; Lee, S.; Park, J.K.; Suh, Y.J.; Leem, J.G.; Shin, J.W. Percutaneous epidural adhesiolysis using inflatable balloon catheter and balloon-less catheter in central lumbar spinal stenosis with neurogenic claudication: a randomized controlled trial. Pain Physician 2018, 21, 593-606.
Kim, S.H.; Choi, W.J.; Suh, J.H.; Jeon, S.R.; Hwang, C.J.; Koh, W.U.; Lee, C.; Leem, J.G.; Lee, S.C.; Shin, J.W. Effects of transforaminal balloon treatment in patients with lumbar foraminal stenosis: a randomized, controlled, double-blind trial. Pain Physician 2013, 16, 213-224.
Shin, J.W. A new approach to neuroplasty. Korean J. Pain 2013, 26, 323-326.
L123: Precisely how long before and after the procedure were patients assessed for pain and ODI? Please add more details.
- Revision Manuscript. Line number 133-142.
Patients’ degree of pain was measured using a standardized 11-point (0–10) Numerical Rating Scale (NRS) and was evaluated by a well-trained physician at baseline and 1 and 3 months after the procedure. The severity of pain was scored from 0 to 10, where “0” represented no pain at all, and “10” represented the worst possible pain. The patients were encouraged to express their feelings regarding the pain.
The Oswestry Disability Index (ODI) was used to assess the patients’ degree of dysfunction. ODI assessments were performed at baseline and 1 and 3 months after the procedure. ODI is a 10-item questionnaire used globally to functionally assess patients with low back pain. However, in this study, we used the 9-item Korean version of ODI, which excludes the assessment of sexual function for cultural reasons.
Statistical Analysis
L135-137: More details are needed in regards to the LMM approach used. For instance, did you run a LMM for repeated measures? What was and how was determined the covariance structure of the model(s) used? What specific variables were tested? All these details are needed when describing your statistics.
- I agree your opinion. We did run a LMM for repeated measures.
Results
L139: "3.1. Participants". This section should be moved up and added to the subheading "2.1 Patients" in the methods. Table 1 should also be moved to the methods for clarity.
- Revision Manuscript. Line number 86-92.
L161-164: Indicate the % difference of the decreases in each time point compared to before the procedure. Additionally, although no significant differences were found between procedures it would be interesting to present effect sizes to better interpret the clinical value and efficiency of each procedure. Therefore, please add the effect sizes for NRS and ODI for each procedure on the different time points.
- Revision Manuscript. Table 3.
Discussion
L182-184: Please restate the aims and summarise results and conclusions in the first paragraph of the discussion for clarity. Also, if you did not find differences between both approaches why are you claiming that the Kambin's triangle approach is clinically efficient? Revise.
- Revision Manuscript. Line number 186-192.
In this present study, there were no significant differences in pain, functional capacity, or the success rate of the procedure between the two groups. However, this is the first randomized trial to demonstrate the clinical efficacy of the Kambin’s triangle approach for percutaneous transforaminal epidural adhesiolysis using an inflatable balloon catheter. The results of this study suggest that the Kambin’s triangle approach is as effective as the conventional safe triangle approach. Therefore, the Kambin’s triangle approach could be an alternative for patients in whom the safe triangle approach for percutaneous transforaminal epidural adhesiolysis is difficult.
L182-194: The first 4 paragraphs of your discussion should added up and summarised in only one (1st) paragraph.
- Revision Manuscript. Line number 186-192.
In this present study, there were no significant differences in pain, functional capacity, or the success rate of the procedure between the two groups. However, this is the first randomized trial to demonstrate the clinical efficacy of the Kambin’s triangle approach for percutaneous transforaminal epidural adhesiolysis using an inflatable balloon catheter. The results of this study suggest that the Kambin’s triangle approach is as effective as the conventional safe triangle approach. Therefore, the Kambin’s triangle approach could be an alternative for patients in whom the safe triangle approach for percutaneous transforaminal epidural adhesiolysis is difficult.
L229: The discussion sections needs to be improved by clearly exploring aims, results and conclusions of this and other studies presented/cited to enable proper comparisons to your results.
- I agree with your opinion. I deleted the paragraph because it didn’t seem necessary for the discussion.
L231-232: "First, the total number of patients included in this study was relatively small, so the study lacked a clear conclusion." Do you mean that the study was underpowered? Did you perform any sample size calculation to avoid this issue?
- We calculated the sample size when planning this study. However, in our opinion, this study in a pilot study that will require further study in the future.
234-235: "functional status". How did you control for this factor between participants of the sample? Additionally, do you consider that the large age range and potential difference on physical level and age between groups may have influenced results of the study? If yes, these points should also be discussed in the limitations paragraph as well.
- Revision Manuscript. Line number 225-227.
Third, a number of confounding variables exist, including the patient’s pain duration and functional status. We suspect that the age distribution of in our study showed a large variation and influenced the results, such as ODI.
L240: "In summary, the Kambin’s triangle approach could be an alternative option...". Revise. Since no significant difference was found between approaches why do you consider that the Kambin's triangle approach could be an alternative option compared to the safe triangle? Presenting non-significant % difference and effect size results between groups may improve such conclusions.
- Traditionally, percutaneous transforaminal epidural adhesiolysis have been performed through safe triangle approach. However there was no significant difference between the safe triangle approach group and the Kambin’s triangle approach group in this study. As a result, we think Kambin’s triangle approach can be a good alternative method for patients who are not easily treated with safe triangle approach.
Reviewer 2 Report
The study is aimed to evaluate the effectiveness and significance of both the Kambin’s triangle approach and the traditional safe triangle approach for percutaneous transforaminal epidural adhesiolysis using an inflatable balloon catheter. The title is “The Kambin’s Triangle Approach for Percutaneous Transforaminal Epidural Adhesiolysis Using an Inflatable Balloon Catheter: A Pilot Study”.
A sample size of the study is relatively small. Some limitations might be occurred. Several factors influence the outcome of the study. Please discuss these. Please review the literature and add more details in the discussion section. What are the new knowledge from this study? Finally, please recommend the readers “How to apply this knowledge in routine clinical practice?”.
Author Response
A sample size of the study is relatively small. Some limitations might be occurred. Several factors influence the outcome of the study. Please discuss these. Please review the literature and add more details in the discussion section. What are the new knowledge from this study? Finally, please recommend the readers “How to apply this knowledge in routine clinical practice?”.
- I agree with your opinion. But this is a pilot study. There is no prior information to estimate sample size. We calculated the number of subjects by referring to the most similar article (pain physician 2013; 16:213-224). Using two group t-test, for 80% power and a two-sided significance level of 0.05, 8 patients were required in each study group, for a total sample size of 16 participants. Assuming 10% dropout, 9 patients were required in each group. Assuming 25% dropout, 10 patients were required in each group. The effect size is estimated at 1.615.
So we referenced the article of Whitehead et al. (Stat Methods Med Res. 2016 Jun; 25(3): 1057–1073.) According to their opinion, for 80% power and smaller standardized effect size, 20 patients were required. Assuming 25% dropout, 13 patients were required in each group.
We also contacted the hospital statistics team. They replied that it was okay to proceed because it was a pilot study.
- Traditionally, percutaneous transforaminal epidural adhesiolysis have been performed through safe triangle approach. However there was no significant difference between the safe triangle approach group and the Kambin’s triangle approach group in this study. As a result, we think Kambin’s triangle approach can be a good alternative method for patients who are not easily treated with safe triangle approach.
Reviewer 3 Report
As a pilot study, it seems methodologically sound.
The manuscript is professionally edited as the authors mentioned.
However, so many small unnecessary paragraphs in the discussion section (please merge many of them and make a few paragraphs).
Fundamentally, the study compared the effectiveness
of the Kambin’s triangle approach versus the safe triangle
approach for PTFA) without control with small samples
as a pilot study. So, the title needs to revise by adding
the word “comparison” between these two approaches. Then study aim also need to be revised, as authors mentioned
the efficacy of Kambin’s triangle approach already demonstrated
then the aim of this study is to compare both approaches. As there were no significant differences between 2 approaches
then rigorous discussion should develop based on previous studies
and its clinical justification. Data on non-significant differences and calculation of effect
size results between groups will be helpful for the reader
or clinicians to understand the fact in a better way
and thus the conclusions can be improved.
Author Response
However, so many small unnecessary paragraphs in the discussion section (please merge many of them and make a few paragraphs).
- Revision Manuscript. Line number 186-192.
I agree with your opinion. I merged the first 4 paragraphs. And I deleted the paragraph. (original manuscript. Line number 226-230.)
Fundamentally, the study compared the effectiveness of the Kambin’s triangle approach versus the safe triangle approach for PTFA) without control with small samples as a pilot study. So, the title needs to revise by adding the word “comparison” between these two approaches.
- Revision Manuscript. Line number 2-5.
I agree with your opinion. I revised it.
The Kambin’s Triangle versus traditional safe triangle approach for Percutaneous Transforaminal Epidural Adhesiolysis Using an Inflatable Balloon Catheter: A Pilot Study
Then study aim also need to be revised, as authors mentioned the efficacy of Kambin’s triangle approach already demonstrated then the aim of this study is to compare both approaches.
- Revision Manuscript. Line number 19-21. 69-71.
I agree with your opinion. I revised it.
As there were no significant differences between 2 approaches then rigorous discussion should develop based on previous studies and its clinical justification. Data on non-significant differences and calculation of effect size results between groups will be helpful for the reader or clinicians to understand the fact in a better way and thus the conclusions can be improved.
- I agree with your opinion. I revised Table 3. The actual effect size was smaller than the expected effect size, result in a non-significant result.
This is a pilot study. There is no prior information to estimate sample size. We calculated the number of subjects by referring to the most similar article (pain physician 2013; 16:213-224). Using two group t-test, for 80% power and a two-sided significance level of 0.05, 8 patients were required in each study group, for a total sample size of 16 participants. Assuming 10% dropout, 9 patients were required in each group. Assuming 25% dropout, 10 patients were required in each group. The effect size is estimated at 1.615.
So we referenced the article of Whitehead et al. (Stat Methods Med Res. 2016 Jun; 25(3): 1057–1073.) According to their opinion, for 80% power and smaller standardized effect size, 20 patients were required. Assuming 25% dropout, 13 patients were required in each group.
We also contacted the hospital statistics team. They replied that it was okay to proceed because it was a pilot study.
Round 2
Reviewer 1 Report
Thank you for considering most of my concerns. The overall quality of the paper has now risen. However, I consider there are still very important revisions to be performed for the paper to be considered suitable for publication. Please see my specific comments below:
Introduction
L53: "significantly higher pain reduction and functional improvement". How much higher? Present the % difference between the two conditions found by this study.
Statistical Analyses
L147: Still need to add information that this was a LMM for repeated measures, as well as how many models were run, and the covariance structure used for each model.
Results
Table 3: Adjust table clearly indicating Z scores and CI, as these are unclear as they are presented.
Discussion
L193-215: These paragraphs still need a better connection to the results of your study, as findings from previous investigations are described but not properly discussed with your findings. Additionally, the effect size results from your results need to be discussed here as well.
L222: "First, the total number of patients included in this 222 study was relatively small, so the study lacked a clear conclusion...". Clearly state that the reason for this is because this was a pilot study, which, therefore, requires follow-up studies in the future to confirm your results.
Author Response
Introduction
L53: "significantly higher pain reduction and functional improvement". How much higher? Present the % difference between the two conditions found by this study.
=> Revision Manuscript. Line number 56-57.
I agree with your opinion. I revised it.
Statistical Analyses
L147: Still need to add information that this was a LMM for repeated measures, as well as how many models were run, and the covariance structure used for each model.
=> Revision Manuscript. Table 4.
The covariance structure we used is unstructure. Since there are no assumptions, the most conservative results are obtained.
Results
Table 3: Adjust table clearly indicating Z scores and CI, as these are unclear as they are presented.
=> Revision Manuscript. Table 3, 4.
I agree with your opinion. I revised it.
Discussion
L193-215: These paragraphs still need a better connection to the results of your study, as findings from previous investigations are described but not properly discussed with your findings. Additionally, the effect size results from your results need to be discussed here as well.
=> Revision Manuscript. Line number 197-243.
I agree with your opinion. I revised it. I further reviewed the literature and added more details in the discussion section.
L222: "First, the total number of patients included in this 222 study was relatively small, so the study lacked a clear conclusion...". Clearly state that the reason for this is because this was a pilot study, which, therefore, requires follow-up studies in the future to confirm your results.
=> Revision Manuscript. Line number 251-252, 261-263, 265-267.
I agree with your opinion. I revised it.
Reviewer 2 Report
The study is aimed to evaluate the safety and efficacy of the Kambin’s triangle approach of percutaneous transforaminal epidural adhesiolysis using an inflatable balloon catheter and to compare this approach to thentraditional safe triangle approach. The title is “Kambin’s Triangle Approach versus Traditional Safe Triangle Approach for Percutaneous Transforaminal Epidural Adhesiolysis Using an Inflatable Balloon Catheter: A Pilot Study”.
1. A sample size of the study is relatively small.
2. Several factors influence the outcome of the study. Please discuss these.
3. Please review the literature and add more details in the discussion section.
4. Please add more details of the limitations of the study.
5. What are the new knowledge from this study?
6. Finally, please recommend the readers “How to apply this knowledge in routine clinical practice?”.
Author Response
A sample size of the study is relatively small.
=> Revision Manuscript. Line number 251-252, 261-263, 265-267.
I agree with your opinion. That is the weakest point of our study. As we replied last time, we think it is appropriate when we combine Whitehead et al.’s opinion with our hospital statistics team. However, it is clear that it should be proved once again by follow-up studies.
Several factors influence the outcome of the study. Please discuss these.
=> Revision Manuscript. Line number 103-104, 254-255, 258-259.
I agree with your opinion. I revised it.
Please review the literature and add more details in the discussion section.
=> Revision Manuscript. Line number 197-218, 236-243, 246-248.
I agree with your opinion. I revised it. I further reviewed the literature and added more details in the discussion section.
Please add more details of the limitations of the study.
=> Revision Manuscript. Line number 251-260.
I agree with your opinion. I revised it.
What are the new knowledge from this study?
=> Revision Manuscript. Line number 156-157, 167-169, 180-182, 190-196, 264-265.
Finally, please recommend the readers “How to apply this knowledge in routine clinical practice?”.
=> Revision Manuscript. Line number 238-243, 264-265.